# How frictional slip evolves

Songlin Shi ®[1], Meng Wang ®[1], Yonatan Poles ®[1] & Jay Fineberg ®[1] ✉

Earthquake-like ruptures break the contacts that form the frictional interface separating contacting bodies and mediate the onset of frictional motion (stick-slip). The slip (motion) of the interface immediately resulting from the rupture that initiates each stick-slip event is generally much smaller than the total slip logged over the duration of the event. Slip after the onset of friction is generally attributed to continuous motion globally attributed to 'dynamic friction'. Here we show, by means of direct measurements of real contact area and slip at the frictional interface, that sequences of myriad hitherto invisible, secondary ruptures are triggered immediately in the wake of each initial rupture. Each secondary rupture generates incremental slip that, when not resolved, may appear as steady sliding of the interface. Each slip increment is linked, via fracture mechanics, to corresponding variations of contact area and local strain. Only by accounting for the contributions of these secondary ruptures can the accumulated interface slip be described. These results have important ramifications both to our fundamental understanding of frictional motion as well as to the essential role of aftershocks within natural faults in generating earthquake-mediated slip.

Slip is the major vehicle for frictional dissipation[1–6]. What, then, determines the 'slip budget', which we define as the slip distribution in space-time? The slip budget is intimately related to the energy dissipation mechanisms associated with frictional sliding[7,8], natural earthquakes[2,9–13] and related phenomena such as seismic estimates of fracture energy[14–16]. Understanding the details of the slip budget is, therefore, a key ingredient to unraveling the complex physical processes that are embodied in the 'everyday' phenomenon that we call 'friction'.

How does frictional slip take place? Experiments have shown that slip initiates via the nucleation of rapid interface ruptures[3–5,17–19], closely akin to earthquakes[14,15,20,21]. These ruptures are shear cracks[4,18,19] that propagate along the frictional interface that separates contacting elastic bodies and break the spatially extended ensemble of contacts that defines the interface. Slip immediately ensues at each point traversed by a rupture[5]. One might expect that, once the initial rupture 'breaks' the interface, slip of the entire interface could occur via continuous sliding that would proceed unimpeded until the applied shear stress within the sample is released. Surprisingly, direct measurements have shown that the initial rapid slip in the immediate wake of rapid rupture fronts is both finite and quite small (<20 μm), while the total slip of each macroscopic stick-slip event is 1–2 orders of magnitude greater[5,12,14,15,22–26]. What physical mechanism governs the total slip?

In our experiments we perform a detailed characterization of all of the interface rupture dynamics from the onset to completion of interface slip within each stick-slip event. We will show that numerous, nearly 'invisible' secondary interface ruptures propagate within a single stick-slip event. Concurrent slip measurements on the frictional interface will yield a nearly complete picture of the slip budget. While the stress and contact variations caused by each of these 'secondary' ruptures is extremely small, their combined slip is significant; nearly all of the slip within the interface is directly generated by these myriad, and nearly undetectable, secondary rupture events.

## Results

Our experimental system[4,27] (Fig. 1a, b) consists of two polyvinyl chloride (PVC) blocks that are pressed together to construct a frictional interface of length $x = 200$ mm and width $z = 5.6$ mm. The longitudinal, $C_L$, shear, $C_S$, and Rayleigh $C_R$ waves speeds in PVC are, respectively 1886, 1067 and 983 ms$^{-1}$. The blocks were first compressed by applying a normal force $F_N = 4300$ N (3.84 MPa nominal stress) in the $y$ direction. The shear force, $F_S$, was then increased slowly until a stick-slip sequence initiated. Upon initiation, we continuously

[1]The Racah Institute of Physics, The Hebrew University of Jerusalem, Givat Ram, Jerusalem 91904, Israel. ✉e-mail: jay@mail.huji.ac.il

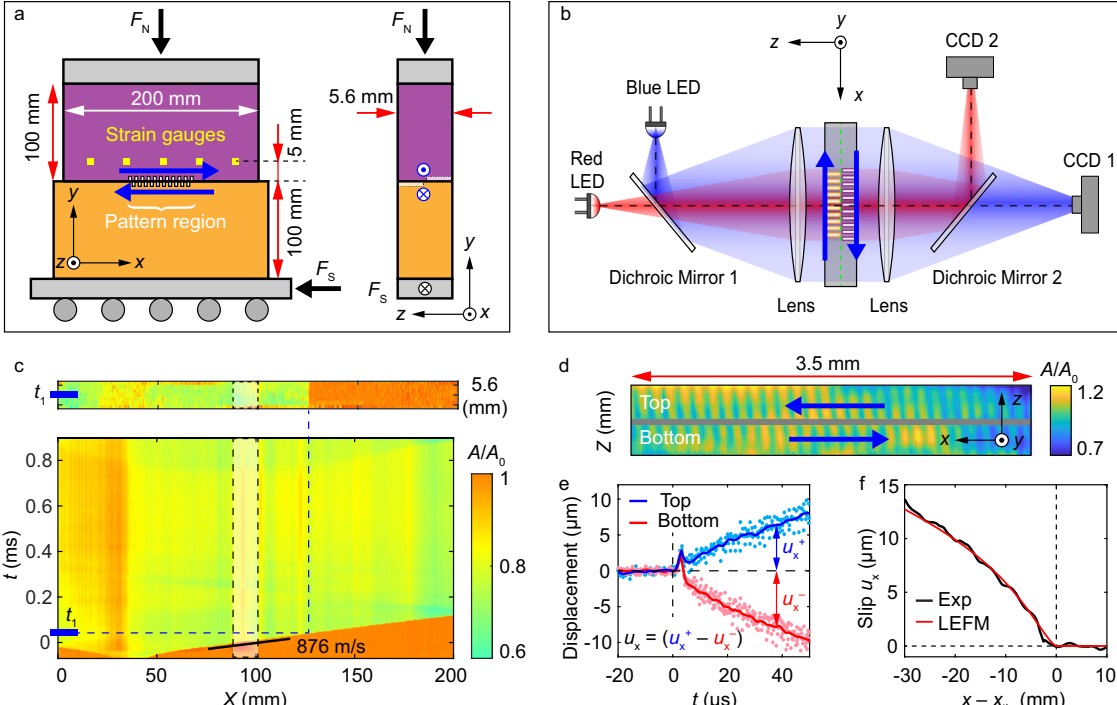

**Fig. 1 | Measuring contact area, strain and slip. a** PVC blocks of thickness $W = 5.6$ mm are compressed under a normal force, $F_N$. Shear force, $F_S$, is applied to the lower block. Five miniature rosette strain gauges (green squares), 5 mm above the interface, are equidistantly distributed in $x$. Grooves are engraved (spatial period 120 µm) between $90 < x < 110$ mm at $0 < z < W/2$ (top block) and $W/2 < z < W$ (bottom block). **b** Optical system for real contact area, $A(x, z, t)$, and slip measurements. The interface is illuminated by blue and red LED light sheets, which are separated by dichroic mirrors. Each light sheet undergoes total internal reflection at the PVC/air interface. Transmitted blue (red) light, captured by fast camera, CCD1 (CCD2) measures $A(x, z, t)$ over the entire (grooved section of the) interface. CCD1 provides rupture dynamics via the space-time variations of $A(x, z, t)$. CCD2 monitors simultaneous displacements of both sets of grooves.

**c** Typical $A(x, z, t)$ measurement at the onset of a stick-slip event. Upper panel: a 2D snapshot of $A(x, z, t_1)/A(x, z, t_0)$, where $t_0 = 0$ is a reference time immediately prior to the event and $t_1 \sim 40$ µs after the onset of the initial interface rupture. Lower panel: $A(x, t)/A(x, t_0) = \langle A(x, z, t)/A(x, z, t_0) \rangle_z$ of this event. Rupture fronts are points where $A(x, t)/A(x, t_0)$ falls from 1. **d** CCD2: Images of grooved regions. The gray line separates grooves engraved on the top and bottom blocks. Blue arrows indicate their relative motion. **e** Measured displacements (see Methods) at the interface of the top (blue, $u_x^+$) and bottom (red, $u_x^-$) blocks when the rupture in (**c**) traversed the grooved section. Lines: $u_x^+$ and $u_x^-$ smoothed over 1 µs. **f** Black line: slip, $u_x = u_x^+ - u_x^-$, at the interface as a function of the distance from the crack tip, $x_{tip}$. Red line: the slip profile predicted by the LEFM, with fitted fracture energy $\Gamma_1 = 1.3$ J m$^{-2}$.

and concurrently monitored: each strain component at 5 locations at a 1.16 MHz rate, the real contact area $A(x, z, t)$ along the entire interface at 580,000 frames/sec with an $x \times z$ resolution of $1280 \times 8$ pixels, and the displacements $u_x^+(x, t)$ and $u_x^-(x, t)$ of the upper and lower blocks at 30 spatial points within the central 3.6 mm of the interface. These displacements were measured by correlating the profiles of grooves engraved on both sides of the interface (Methods and Fig. 1d). These micro-scale grooves have negligible influence on the rupture propagation, resulting strain fields and the value of the fracture energy, as demonstrated in Supplementary Figs. 1, 2 and Methods. Slip $u_x(x, t) = u_x^+(x, t) - u_x^-(x, t)$, was obtained at each point at a rate of 560,000/sec with a 0.5 µm resolution (Methods).

In Fig. 1c–f, we present typical measurements of $A(x, z, t)/A(x, z, t_0)$, $u_x^-(x, t)$, $u_x^+(x, t)$, and $u_x(x, t)$ for a right-propagating sub-Rayleigh rupture (velocity, $C_f = 0.89C_R$ at $x = 100$ mm) that initiated at $x = 40$ mm and traversed the region $90 < x < 100$ mm where slip was measured. $A(x, z, t_0)$ denotes the initial contact area at a time $t_0$ prior to rupture initiation. Each rupture front reduces $A(x, z, t)$ by partially detaching the interconnecting contacts that form the interface. $A(x, z, t)/A(x, z, t_0)$ provides the location, $x_{tip}$ and front velocity $C_f$ of each rupture front as it traverses the interface. At each spatial point traversed by a rupture, slip takes place.

Both $u_x^+(x, t)$, and $u_x^-(x, t)$ for the first rupture in Fig. 1c are presented in Fig. 1e. As the rupture tip traverses each point, surprisingly both $u_x^+$ and $u_x^-$ are first displaced (for about 5 µs) in the same direction. While this effect precedes rapid motion[5], it does not

contribute to slip, $u_x = u_x^+ - u_x^-$, as $u_x^+$ and $u_x^-$ cancel each other. Beyond this interval, slip ($u_x > 0$) initiates via roughly antisymmetric displacement of the two blocks (Fig. 1e), although perfect antisymmetry of $u_x^\pm$ is not always observed. Figure 1f reveals that the $u_x(x - x_{tip})$ structure generated by the sub-Rayleigh rupture compares well with the slip profile calculated with Linear Elastic Fracture Mechanic (LEFM) with a cohesive model[28] (Methods, Supplementary Fig. 1a, b).

The first interface rupture produces the relatively large contact area drop, $\Delta A/A_0 \approx 0.26$ shown in Fig. 1c. Following this initial drop, subsequent weaker and considerably less distinct contact area reductions occur (e.g., beyond $t_1$ in Fig. 1c). Ostensibly, the apparent 'constant' value of $A/A_0$ for $t > t_1$ could suggest a continuously sliding process; i.e., 'dynamic friction'. A closer look (Fig. 2a), via (Methods) the normalized differential contact area $\overline{\partial_t A(x, t)}$ at each spatial location, $x$, reveals a different picture. Within the first 3 ms of slip in Fig. 1c, numerous sequential secondary ruptures[6,16,29–32] take place after the main rupture. These secondary ruptures, revealed by $\overline{\partial_t A(x, t)}$, are nearly invisible in the contact area measurements, as resulting $\Delta A/A_0$ are extremely small (Supplementary Table 1).

Details of secondary ruptures are presented in Fig. 2a, b. Red dashed lines denote the times when each secondary rupture arrived at the region ($x \sim 90$ mm) where slip was measured. The corresponding slip and $\Delta A$ variations within each time interval, are presented in Fig. 2c, d. Each sequential rupture causes a drop in contact area accompanied by slip. This sequence of secondary ruptures only ceases

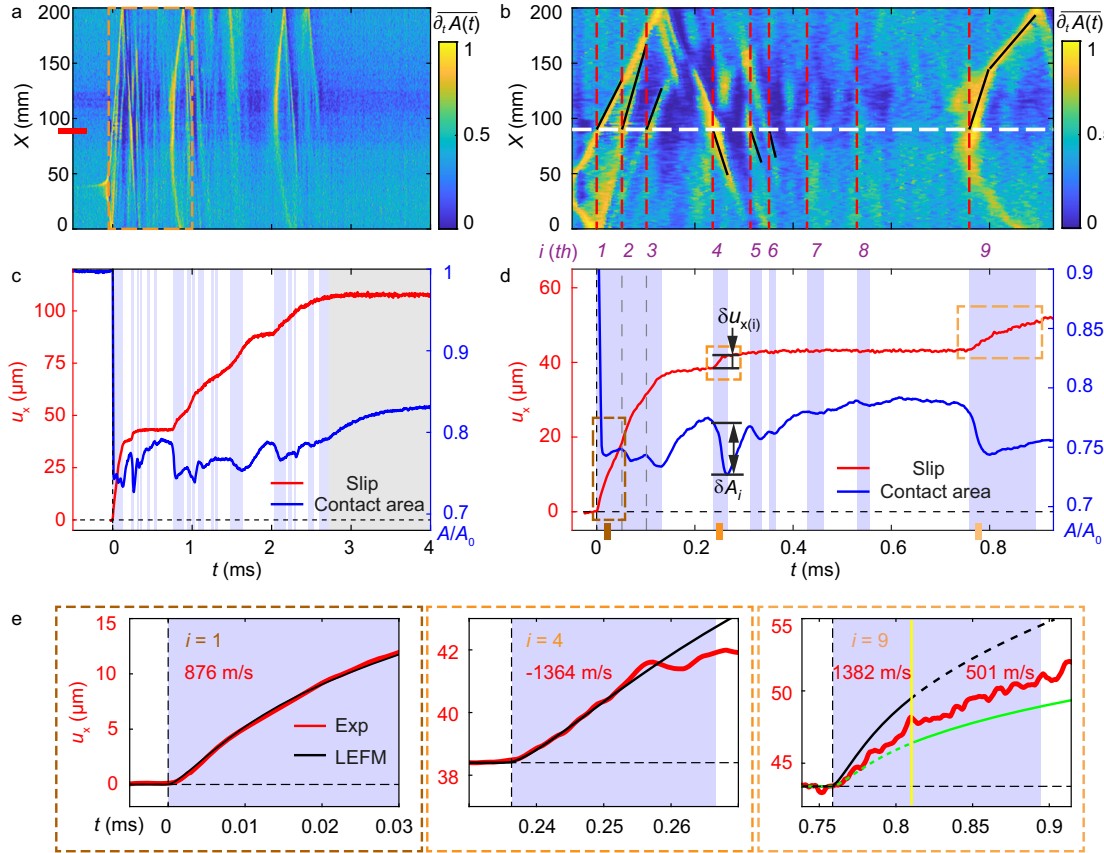

**Fig. 2 | The evolution of secondary ruptures. a** Normalized differentiation $\overline{\partial_t A}(x,t)$ reveals numerous sequenced ruptures occurring in the 3 ms following the initial rupture. **b** The first 1 ms of (**a**). Vertical dashed lines labeled by integers denote sequential ruptures as they pass the slip measurement location. **c**, **d** The detailed evolution of the contact area (blue line) and the slip (red line) in (**a**, **b**), respectively. The light purple backgrounds denote the temporal windows over which the labeled ruptures continued to propagate within the interface. $\delta u_i$ and $\delta A_i$ denote the slip and contact area drop precipitated by $i^{th}$ rupture. The slip measured

at the $X$ position denoted in (**a**) (red mark) and (**b**) (white dashed line). The slopes of the black lines in (**b**) denote the local rupture velocities as ruptures cross the measurement section. **e** Three typical ($i = 1, 4, 9$) slip profiles (red) are compared to LEFM predictions (black). $i = 1$ is sub-Rayleigh, while $i = 4$ is supershear. The $i = 9$ rupture is supershear until the yellow line, it then transitions to sub-Rayleigh. Black and green curves are LEFM predictions within the supershear and sub-Rayleigh ranges, respectively.

after $t \sim 3$ ms, when the contacting blocks become locked and $A(x, z, t)$ commences to reheal (or 'age') as $A(t)$ increases logarithmically in time[5,33–35].

The slip at each spatial point is not smooth, but, instead, increases in discrete increments; each secondary rupture initiates augmented slip at each spatial location, immediately upon the rupture's passing. The slip initiated by each rupture continues until information (via shear waves) that the rupture has either arrested or traversed the entire interface reaches the measurement point. Some ruptures, e.g., between 0.5 and 0.8 ms, while detectable, are so weak that they are not explicitly marked. These extremely weak ruptures still have a marked effect; they prevent aging of $A(x, t)$ and give rise to apparent 'noise' together with a very slight increase of slip in the $u_x(t)$ measurements. The total slip reaches 120 μm, an order of magnitude greater than the 'large' slip generated by the initial rupture (Fig. 2c, d).

For convenience, in Fig. 2 we label the more prominent secondary ruptures by their chronological order. The durations of the first three ruptures overlap, as they propagate simultaneously within the interface. The 2nd and 3rd ruptures are formed by reflections from the $x = 0$ mm boundary. The 2nd rupture is a supershear rupture ($v > C_S$)[36–38], while the 3rd rupture propagates close to the Rayleigh wave speed, $C_R$. We find that the majority of the observed sequential ruptures are supershear, in contrast to other observations[16,29]. This difference may be attributed to significant differences in stored elastic energy, prior to nucleation of the initial ruptures. In our experiment,

initial ruptures attain speeds, $v > 0.8C_R$, whereas $v \sim 0.1C_R$ in previous observations of secondary ruptures[16,29]. Rapid initial ruptures (analogous to an earthquake's 'main shock') tend to trigger numerous secondary supershear ruptures; such supershear aftershocks have not yet been frequently identified in natural earthquakes[39].

Generally, the mode of each secondary rupture (sub-Rayleigh vs. supershear) tends to be retained during propagation. In some cases, however, a rupture can transition from sub-Rayleigh to supershear[36,37,40,41], or vice versa[41]. An example is presented in the $i = 9$ rupture (Fig. 2e, right panel) that nucleates as supershear before transitioning to sub-Rayleigh. Such transitions may occur when ruptures encounter barriers or obstacles[3,41].

Often, in our system, additional, 'bunched' rupture sequences occur, such as those occurring during $t = 0.8–1.2$ and $2–2.4$ ms in Fig. 2. Each of these proceeds as in Fig. 2a. The intervals between these slip sequence clusters are consistent with the ~1 ms time scale, defined by our system's resonant response time (Methods, Supplementary Figs. 3, 4). The weak forcing caused by any small resonant 'backlash' is sufficient to trigger additional rupture sequences.

The fracture energy of a frictional interface is a dynamic quantity. The fracture energy of the $i$th rupture, $\Gamma_i$ is determined by both the contact area which increases with the normal stress, $\sigma_{yy}$, and the contacting time $\Delta t$[42–44]. The 'invisible' character of secondary ruptures, stems from the fact that the intervals, $\Delta t_i$, since the previous rupture are generally quite small (often less than 10–20 μs, as in the 5th and 6th

ruptures in Fig. 2b, d) and leave little time for $A(t)$ to reheal. This contrasts with the significant contact aging[5,33–35] over the interval $\Delta t_1$ before the initial rupture. Since $\sigma_{yy}$ is constant, we can assume that the fracture energy of the $i$th rupture, $\Gamma_i$, is proportional to the contact area drop $\delta A_i$ that they precipitate (see Fig. 2d); therefore $\Gamma_i = \Gamma_1 \cdot (\delta A_i / \delta A_1)$ (Methods). As Fig. 2d demonstrates, $\delta A_i$ are generally quite small, often below 5% of $\delta A_1$ (Supplementary Table 1). These minute $\delta A_i$ explain the extreme velocities of the secondary ruptures, as when $\Gamma_i \to 0$, any small (but finite) driving stress will cause a rupture to immediately accelerate to extreme values.

We now examine three specific ruptures (1st, 4th, and 9th) that produce finite slip steps. In Fig. 2e, we show that LEFM[28,45] successfully describes the slip profiles resulting from both sub-Rayleigh (1st) and supershear (4th) ruptures (Methods, Supplementary Fig. 5). Secondary ruptures are not, necessarily, well-defined rupture modes, as in the 9th rupture that transitioned from supershear to sub-Rayleigh modes. Even here, LEFM predictions are close to measured values. Such transitions can (and should) affect the slip profile at a given point at the interface. As no theoretical framework exists that predicts the slip profile in the wake of a rupture that transitions to a different rupture mode (as in the 9th rupture), we assume that superposition of the slip profiles of the different rupture modes can be used.

In Fig. 3a, we present strain gauge measurements that accompany the secondary ruptures described in Fig. 2. The largest $\varepsilon_{ij}$ component, $\varepsilon_{xx}$, is presented. $\varepsilon_{ij}(x - x_{\text{tip}})$ of the first ruptures in each stick-slip sequence are reproduced well by LEFM (Supplementary Fig. 1c). Secondary ruptures often propagate within the tail of previous ruptures, so secondary ruptures may dynamically vary the residual stresses in both space and time along interfaces. This leads to dynamically induced stress inhomogeneities within interfaces. Both this effectively variable residual stress and the very small stress changes induced by secondary ruptures introduce challenges in accurately comparing the spatiotemporal strain profile to theory. Despite these difficulties, we can utilize the peak-to-peak strain variations, $\delta \varepsilon_{ij}^{\text{osc}}$, that are excited in close proximity to passing rupture tips, to perform a rough

comparison to LEFM predictions[4]. Examples of $\delta \varepsilon_{xx}^{\text{osc}}$ are denoted in Fig. 3a. It is clear that the signal levels of $\delta \varepsilon_{ij}^{\text{osc}}$, while appearing 'noisy', are actually far about our measurement noise (magenta dashed lines in Fig. 3b, c). These fluctuations are entirely due to the passage of secondary ruptures.

Figure 3b compares the peak-peak strain oscillations, $\delta \varepsilon_{ij}^{\text{osc}}$, of all strain components as a function of the order of the appearance of the secondary ruptures, as denoted in Fig. 2c. Figure 3c presents the corresponding drop $\delta A_i$ in $A / A_0$ for $i < 22$. As the mean $\sigma_{yy}$ is constant, the $\delta A_i$ represent the relative fracture energy $\delta \Gamma_i$ of each rupture. ($\delta \Gamma_i$ are set by both the healing interval, $\Delta t_i$, between sequential ruptures and their velocity[43]). In the $\Delta t_i \approx 20$–100 μs between secondary ruptures, the recovery of $A(x, t)$ is nearly imperceptible, $\delta A(x, t) < 1\%$[5]. This is barely within our 0.2–0.5% measurement resolution. In Fig. 3d we compare all of the measured $\delta \varepsilon_{ij}^{\text{osc}}$ with their values predicted by LEFM (for both sub-Rayleigh or supershear solutions) supplemented by a simple cohesive zone model[28] (see Methods) whose sole input is $\Gamma_i$. The measured and predicted strain oscillations agree quite well; LEFM quantitatively describes the behavior of secondary ruptures. To perform this comparison, we only considered isolated ruptures whose velocities could be quantitatively determined (Supplementary Table 1). Weak ruptures with $C_f \to C_R$ were excluded, as in this range any small uncertainty in $C_f$ could result in significant uncertainties in predicted strains.

We define an 'event' as containing both the initial rupture (indexed by '1') together with all of its subsequent accompanying secondary ruptures. We define the total slip of an event, $u_{\text{tot}}$ as the integrated slip of all of these ruptures. Whereas the slip of the initial rupture is typically $u_1 \sim 10 - 20$ μm we find that $u_{\text{tot}} \sim 100 - 200$ μm (e.g., Figs. 1, 2). How much of $u_{\text{tot}}$ is directly due to the secondary ruptures? In Fig. 4a we present the evolution of $u_x$ from the time of the initial rupture until the cessation of slip, in two representative events. As we would expect, $u_{\text{tot}}$ is linearly related to the overall shear stress drop, $\Delta \sigma_{xy}$ (Fig. 4a – inset). This observation is consistent with the linear relation of the stress drop and slip in earthquakes[13]. On the other

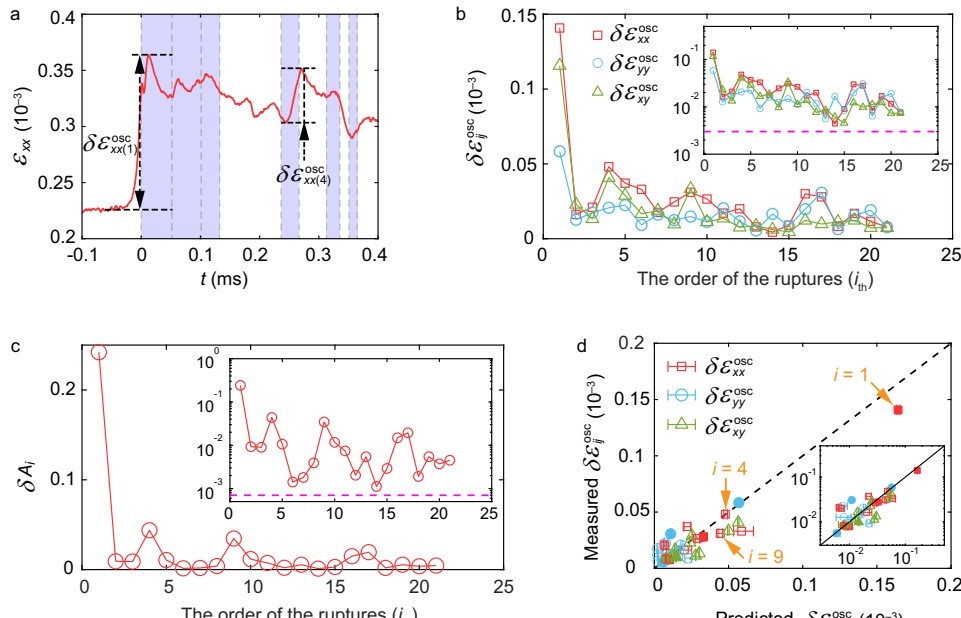

**Fig. 3 | Strain and contact area drops due to secondary ruptures are described by LEFM. a** Strain gage signals $\varepsilon_{xx}$. Noted are strain oscillation amplitudes, $\delta \varepsilon_{xx}^{\text{osc}}$, from the 1st and 4th ruptures of the sequence in Fig. 2. Shading denotes, as in Fig. 2, rupture durations. **b** The strain oscillation amplitudes of three components ($\delta \varepsilon_{xx}^{\text{osc}}$, $\delta \varepsilon_{yy}^{\text{osc}}$, $\delta \varepsilon_{xy}^{\text{osc}}$) due to a rupture sequence, as a function of their order of appearance. **c** The contact area drops $\delta A_i$ of the sequence presented in (**a**, **b**). Insets in (**b**, **c**) are

plotted in logarithmic y-coordinates; magenta horizontal lines are measurement noise levels. **d** Comparison of measured $\delta \varepsilon_{ij}^{\text{osc}}$ with LEFM predictions, using the fracture energy $\Gamma_i = \Gamma_1 \cdot [\delta A_i / \delta A_1]$ as input. $\Gamma_1 = 1.3\,\text{Jm}^{-2}$. Filled and open symbols represent sub-Rayleigh and supershear ruptures, respectively. The $\delta \varepsilon_{xx}^{\text{osc}}$ of the ruptures $i = 1, 4, 9$ in Fig. 2e are marked by arrows. Inset: logarithmic coordinates.

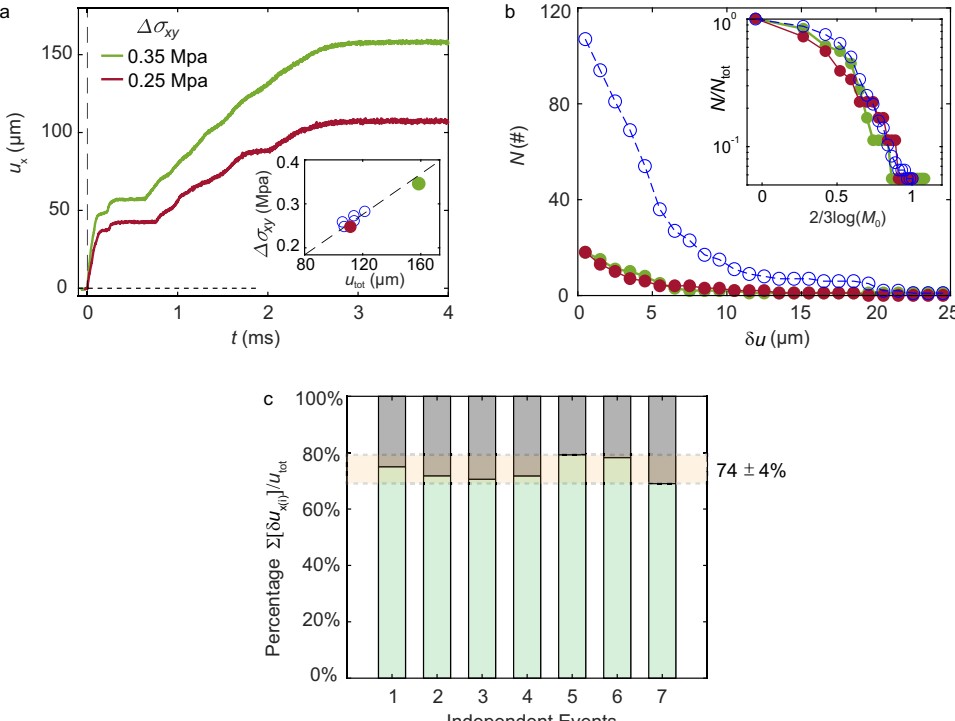

**Fig. 4 | Slip Budget. a** The slip profiles of two typical events (with different sequenced ruptures) under different global stress drops, $\Delta\sigma_{xy}$, of 0.25 and 0.35 Mpa. The global stress drop is defined as the difference between the initial stress before the first rupture occurs and the averaged stress after the first rupture until slip cessation. Higher global stress drops $\Delta\sigma_{xy}$ correspond to larger total slips $u_{tot}$. Inset: global stress drop and total slip from seven events, $u_{tot} \propto \Delta\sigma_{xy}$, the reference dashed line passes through (0,0). **b** Statistics of the number of sequenced ruptures whose slip exceeds $\delta u_x$. Blue open symbols are the statistics from 7 different events. Red and green symbols are the corresponding events from Fig. 4a. Inset: statistics of corresponding (normalized) magnitudes. **c** Percentage of total slip that is directly due to ruptures (see text) in 7 different events (from different experiments); ruptures traversing the measurement region directly account for at least 74% ± 4% of the total slip for each event.

hand, within each event, slip distributions of both the primary and secondary ruptures can (and will) vary considerably, as in the examples presented. In Fig. 4b, the number of ruptures whose slip exceeds $\delta u$, is presented for both the events presented in Fig. 4a (green and red symbols) as well as (blue symbols) the cumulative results of 7 ruptures (each from a different experiment). These distributions (inset - Fig. 4b) collapse onto one another, when both normalized and scaled by their approximate seismic moment $M_0 = G \cdot S \cdot \delta u_i$. $G$ and $S$ are, respectively, the shear modulus and total interface area while $\delta u_i$ is the slip of each secondary rupture[13] (Methods).

The $M_0$ of the secondary ruptures fall within 1–50 N·m. While most of these ruptures traversed the entire length of our experimental 'fault', their slip, $\delta u_i$, varies greatly. Slip variation is the main reason for the broad distribution of $M_0$ in Fig. 4b; ruptures precipitating large slip are far less frequent than those with smaller slip. The scaling in Fig. 4b is similar to the Gutenberg-Richter (G-R) law[9] that describes the moment distribution for natural earthquakes; power law behavior is observed on the right side of the curve with a roll-off (top-left corner) for small equivalent magnitudes[13]. The roll-off in Fig. 4b could be due (as surmised for earthquakes[13]) to our lack of resolution in detecting ruptures that generate extremely small slips. Our results may suggest that the main reason for the spread of $M_0$ is not variations of the spatial extent of the different events, since the ruptures observed here all roughly span the interface length. Instead, $M_0$ is governed by $\delta u_i$. $\delta u_i$ are mainly controlled by the fracture energy, $\Gamma_i$, of each rupture, which, itself is governed by the healing time $\Delta t_i$. This could also be true for aftershock moments within natural faults[44].

Let us now consider the 'slip budget' contributed by dynamic ruptures. To what extent is the total measured slip due to the nearly 'invisible' secondary ruptures? In Fig. 4c we compute the percent of

$u_{tot}$ contributed solely by dynamic secondary ruptures in 7 typical experiments, including only ruptures whose stress drops exceeded twice the measurement noise. Surprisingly, we find that main and these secondary ruptures account for approximately 74% of $u_{tot}$. Since we are certainly missing the smallest ruptures, Fig. 4c, provides a lower bound for the slip budget.

## Discussion

We have presented a comprehensive study of the origins of frictional slip. The specific events described in Figs. 1–4 are wholly representative of the numerous stick-slip events that we have analyzed. In this rather standard block-on-block frictional motion, slip is entirely dominated by myriad, seemingly 'invisible' secondary frictional ruptures. In all cases that were amenable to detailed analysis (i.e., sufficiently large signals as well as secondary ruptures that were isolated, in time, from other ruptures), we have shown that these ruptures can be effectively described using LEFM supplemented by a simple cohesive zone model. The apparent 'invisibility' of the secondary ruptures, is due to the, often minute, healing intervals available to sequential ruptures; as aging is approximately logarithmic in time[33–35,44]. This minute recovery of $\delta A_i \sim \Gamma_i$ gives rise to correspondingly small variations in both the stress changes, $\delta\varepsilon_{ij}$, and slip, $\delta u_i$. While all of these quantities are quite small, since $\Gamma_i \to 0$ the velocities of secondary ruptures are generally extreme (the majority are supershear). At such small $\Gamma$ secondary ruptures can be triggered by either (real) noise in the system or, simply, via reflected ruptures from the horizontal boundaries. The rupture-generated signals excite 'apparent' measurement noise that is significantly above our measurement accuracy. These effects would generally be hidden in global measurements and would appear as steady sliding. To detect them it

is necessary to have sufficient temporal accuracy (μs time scales) together with the ability to observe and correlate all of these, seemingly, insignificant variations in $A, \varepsilon$, and $\delta u$. As we have shown in both Figs. 2e, 3d, these miniscule and seemingly 'disparate' signals are indeed correlated via LEFM.

These findings provide a conceptually different view of the evolution of sliding that, macroscopically, is generally attributed to 'dynamic friction' at the transition from 'static friction', occurring after the initial rupture. Global measurements that average over spatial and significant temporal scales present a mean-field perspective of the average slip. These would manifest as the smooth curve assumed in empirical friction laws[7,23]. Such apparently continuous macroscopic motion that is, instead, driven by a sequence of small discrete events is also observed in the peeling dynamics of adhesive tape[46]. While the details are different than in the frictional system, the apparent slow and continuous macroscopic motion obtained when peeling an adhesive strip is actually produced by a discrete succession of nearly imperceptible rapid fronts, each producing a small and discrete amount of slip.

Now that the basic physics of secondary frictional ruptures are understood, we speculate that it may be possible to utilize this knowledge to control the amount of frictional slip that takes place in a given event. In principle, external excitation of numerous secondary ruptures should enhance overall frictional slip, while the insertion of barriers to rupture[3] could be a means to limit slip by arresting (trapping) secondary ruptures.

When we consider the energy dissipated by this process, the fracture energy released at the rupture tip, $\sum \Gamma_i \sim \sum (\tau_p - \tau_{res})_i \cdot u_i$ (Methods), is a relatively small part of the total dissipation. Here $\tau_p$ and $\tau_{res}$ are the ($i$ · dependent) peak and residual stresses. The majority of the frictional dissipation $\sim u_{tot} \cdot \tau_{res}$, is also related to the fracture process through $u_{tot}$, as most of $u_{tot}$ is directly attributed (Fig. 4) to the rupture process. The dissipation and 'damage' (damage here is measured by the variation of the local contact area) imposed by the primary and secondary rupture processes are non-uniform along the interface. These ruptures, with their accompanying slip, introduce non-uniform residual strengths along the interface. This non-uniformity is retained upon the cessation of slip. While the interface strength at each point will heal via the approximate logarithmic aging in time[5] of $A$ (Supplementary Fig. 6), any inhomogeneity of $A$, upon the cessation of slip, will be retained by this healing process. $A(x, t)$ depends on the values of the residual strength at the cessation of slip, local stress levels and the time since slip cessation at each spatial point.

The complex framework of inhomogeneous and coexisting sequenced ruptures are statistically observed as aftershocks of large earthquakes within natural faults, e.g., G-R law (subfigure of Fig. 4b). These myriad events are so numerous (and so small) that they are individually invisible and are generally only represented by statistical tools (e.g., Gutenberg-Richter analysis). The secondary ruptures that we observe are aftershocks, each is part of a swarm of rupture events in the immediate aftermath of any 'main' stick-slip event. These rupture sequences could provide a means to relate observed correlations between geodetically observed afterslip and seismically observed aftershocks[10,11]. These experiments provide valuable insights to complex fault processes, as detailed observation of individual ruptures within natural faults is currently extremely challenging.

We demonstrated that nearly invisible secondary ruptures that always follow the first rapid frictional ruptures predominantly shape the slip budget. The repeated breaking and localized healing along the interface, therefore, play a crucial role in governing the evolution of the 'frictional tail' that follows the onset of friction, as well as the relationship between after-slip and aftershocks.

## Methods

### Sample construction
We define $x$, $y$, $z$ as, respectively, the sliding, normal loading, and surface thickness directions. The $x \times y \times z$ dimensions of the top and bottom PVC blocks were $200.0 \times 100.0 \times 5.6$ mm and $220.0 \times 100.0 \times 5.6$ mm. Both blocks were polished to optical flatness with a roughness ~3 μm. The longitudinal wave speed $C_L$ and shear wave speed $C_S$, were measured ultrasonically, yielding values $C_L = 1886 \pm 10$ m s⁻¹(plane stress) and $C_S = 1067 \pm 10$ m s⁻¹. These provide a Rayleigh-wave speed $C_R = 983 \pm 10$ m s⁻¹ and yield dynamic values of the Poisson ratio, $v = 0.36$, and Young's modulus of $E = 4.18$ GPa. The density of the PVC is 1350 kg m⁻³.

### Groove pattern engraving
The motion of grooved patterns on the frictional interface is used to track the slip of the interface. To avoid interference between the patterns on the upper and bottom blocks, groove patterns are only engraved on half of the width of each block; grooves extended over $0 < z < W/2$ and $W/2 < z < W$, for the upper and lower blocks respectively (Fig. 1d). This technique was used previously, when only a single block face was grooved[27]. Grooves were engraved in the center 20 mm (in $x$) of the interface by means of a UV laser cutter, while half of the interface width was masked. Each groove was 60 μm wide and the periodicity between grooves was set to 120 μm. This procedure formed ridges and gaps of equal width, with a gap depth of 10–15 μm. The entire interface was then re-polished to remove any residual material, formed by the laser cutter, at the groove edges.

### Loading system
The loading system was comprehensively described previously[4]. The top block was clamped at its upper edge, while the bottom block was attached to a low friction translational stage. A uniform normal load $F_N = 4300$ N, was applied at the initiation of each experiments and retained to be constant over the duration of each experiment. Once $F_N$ was set, an external shear force $F_S$ was quasi-statically applied to the bottom block in the negative $x$ direction (from the left side of the bottom block, Fig. 1a) with rate 10 μm s⁻¹. This slow loading rate ensures that $F_S$ is constant over the 3–4 ms duration that characterized the onset of each stick-slip event. After the normal load is applied, the time between the different events influences the value of the real area of contact (bare fracture energy), as the interface ages logarithmically with time; a longer aging time indicates a stronger binding interface. The waiting time for each event was controlled to determine the fracture energy of the first rupture in each stick-slip sequence. The fracture energies of secondary ruptures were determined by the interface evolution and healing times before each onset of a secondary slip.

### Resonant timescale of the mechanical system
Like any mechanical system, our experimental system contains mechanical resonances that introduce an inertial timescale into our experiments. After the initial slip takes place, the loading frame itself will undergo slight oscillations, as these resonances are excited. These oscillations induce slight reloading of the system and may induce the excitation of new rupture sequences within the interface, which has been weakened after the passage of the first rupture sequence. As a result, we often observe successive slip sequences that are spaced in time at the system resonances. In Supplementary Fig. 3 we present the first 10 ms of motion in a typical experiment. The antisymmetric motion of the opposing blocks indicates slip ($u = u^+ - u^-$) – which takes place over approximately the first 3 ms. For longer times, symmetric motion of $u^+$ and $u^-$ takes place that indicates the interface is locked with no slip is taking place. Over this time, the system is undergoing coherent resonant oscillations.

In Supplementary Fig. 4, we present the variations of strains of all three components $\varepsilon_{xx}$, $\varepsilon_{yy}$, and $\varepsilon_{xy}$, after striking the system with a steel rod ($\delta$ function excitation) while applying $F_S$ at levels slightly below the onset of stick-slip (under same $F_N$). The resulting Fourier spectra of the strain gauge signals in the center ($x = 100$ mm) of the blocks produces resonant peaks at 1250 Hz and 400 Hz that correspond to oscillation periods of 0.8 ms and 2.5 ms. These oscillation time scales indeed match the temporal separation of rupture sequences in our experiments and also can be seen in the $u^+$, $u^-$ measurements of Supplementary Fig. 1c, in for $t > 3$ ms when the interface is locked.

After the onset of rupture, these stress oscillations can drive additional rupture sequences (see Fig. 2a) that initiate at these resonant time scales. We wish to emphasize that this effect, in no way, takes away from the generality of our results. It, instead, emphasizes that any type of small perturbation will indeed excite rupture sequences (with their resultant slip), once the interface is weakened by its initial rupture.

## Contact area measurements and contact area differentiation

To image the interface under different scales of the field of view, we utilized two separate high-speed cameras (Fig. 1b). The first high-speed camera (CCD1, Phantom v710), was used to visualize changes in the real contact area, $A(x, z, t)$, over the entire interface at $x \times z = 1280 \times 8$ pixel resolution. Each pixel was, therefore, mapped to an area $\Delta x \times \Delta z = 156 \times 700$ μm. $A(x, z, t)$ was measured via a light sheet that illuminated the entire interface at an angle well beyond the angle for total internal reflection between the PVC and air. As a result, transmitted light only passed through the contact points within each pixel, while light impinging non-contacting sections of the interface was reflected away from the interface. The transmitted light is therefore proportional to $A(x, z, t)$ (top panel in Fig. 1c). Signals were often averaged in the width $z$ to provide $A(x, t)$ (bottom panel in Fig. 1c). The camera was operated at a frame rate of 580,000 frames/sec, enabling continuous measurements of $A(x, z, t)$ at temporal intervals of $\Delta t = 1.7$ μs. To avoid interference and polarization effects, the interface was illuminated using an incoherent blue (460 nm) light source, a high-power LED (CBT-120). This camera has an approximate 9 bit resolution in intensity. After averaging in the $z$ direction, a measurement noise level of the intensity of approximately 0.2% was achieved.

To resolve the weaker secondary ruptures, we utilized the differential contact area $\partial_t A(x, t)$ at each spatial location, $x$, along the interface. The normalized differential contact area $\overline{\partial_t A(x, t)}$ (Fig. 2a) is obtained through $\overline{\partial_t A(x, t)} = [\partial_t A(x, t) - \partial A_c]/\Delta(\partial A)$, where $\partial A_c$ and $\Delta(\partial A)$ are, respectively, the local threshold value of the background and the maximum values of the differentiated signal. To reduce noise in the differentiated signal, running averages in time (20 frames) and space (20 pixels) were used – effectively smoothing over an approximate 3 mm interval.

## Interface displacements and slip measurements

The accurate determination of slip precisely at the interface poses a formidable challenge[27], as rapid, high precision, simultaneous measurements of the relative motion of both contacting surfaces that form a frictional interface is required. The second high-speed camera (CCD2, Phantom v1611) utilized the same imaging technique, but instead was focused on the central $(\Delta x, \Delta z) = (10, 5.6)$ mm grooved section of the interface. Here the 560,000 frame/sec provided a continuous measurement of $A(x, z, t)$ every 1.8 μs (Fig. 1d, e). This region was illuminated by the same type of high-power LED (CBT-120) that produced a highly focused red (623 nm) light sheet. To measure the slip of the interface to high precision, an $x \times z = 3.65 \times 5.6$ mm region was imaged with a resolution of $512 \times 32$ pixels. This enabled visualization of 30 grooves with a spatial resolution in $x$ and $z$ directions of 7 μm and 175 μm for each pixel. Our slip measurements provide the mean slip of each $x \times z = 3.65 \times 2.8$ mm half-width section to 0.5 μm

accuracy. The slip was detected as follows. $A(x, z, t)$ was averaged in $z$ to provide $A(x, t)$. Sequential images in time of $A(x, t)$ between each groove were then cross-correlated to provide $u^\pm(x, t)$ for each $x$. We are interested in obtaining the function $u^\pm(x - x_{tip})$. With CCD1 we both measured $C_f$ and determined the tip location, $x_{tip}(t)$. Assuming $C_f$ to be constant within the measurement time, we then converted the temporal measurements of $\delta u(x, t)$ between each groove to estimate $u^\pm(x - x_{tip})$. A constant $C_f$ provides translational invariance of the rupture front. All quantities are the functions of the argument $x - C_f t$ over the measurement interval. In particular, for grooves separated by a distance $d$, $u^\pm(x, t) = u^\pm(x + d, t + d/C_f)$. This enabled us to align each of the groove measurements to a reference point in the center of the measurement section and average over these independent measurements to obtain $u^\pm(x - x_{tip})$. The relative slip, $u = u^+ - u^-$ was thereby obtained with a resolution of better than 0.5 μm as presented in Fig. 1e (blue $u^+$ and red $u^-$ curve). We note that the grooves produce a negligible effect on the rupture propagation, as shown in Supplementary Fig. 2a. Furthermore, the strain measurements and their correspondence to LEFM at $x = 100$ mm (with grooves, Supplementary Fig. 1c) and $x = 140$ mm (without grooves, Supplementary Fig. 2b) show that both the grooved and non-grooved locations have the same fracture energy. While the grooves redistribute the contact area at the micro-scale within the interface (e.g., the local contact density increases at groove edges and is zero within the grooves), this does not influence the macro-scale rupture properties (i.e., the nominal stress distribution millimeters away from the interface or the propagation speeds).

We note in the main text that, during the first 5 μs, both blocks show the same positive displacement. One possible explanation for these displacements is that they are caused by local material rotational deformation under the action of amplified shear stresses by the approaching singularity.

## Determining the propagation speed of secondary ruptures

Differentiation of the contact area, $\overline{\partial_t A(x, t)}$, highlights the contact area changes induced by weak secondary ruptures. To detect the propagation velocities, $C_f$, of these weak fronts, we first convert the $\overline{\partial_t A(x, t)}$ maps (e.g., Fig. 2a) to contour plots. These contours produce ridges along the $x, t$ paths that the rupture fronts trace. $C_f$ is determined by performing a linear fit to this ridge. Some examples of this procedure are presented in Supplementary Fig. 5. In these examples we illustrate the $i = $ 1st, 4th, and 9th rupture fronts presented in Fig. 2e. Secondary rupture speeds of the secondary rupture sequence described in Fig. 2 are listed in Supplementary Table 1. A portion of the secondary ruptures are so weak that we can not reliably detect their speeds. These rupture speeds are left undetermined in Supplementary Table 1.

## Fracture energy

The initial fracture energy $\Gamma_1$ is determined by the interface contact quality, which is influenced by the normal loading force, the static aging time from imposition of $F_N$ until the initiation of the first rupture, and the surface's roughness[42,43]. In our experiments, $\Gamma_1$ was obtained from fitting the slip profiles to LEFM predictions[28,45]. The value corresponds well to the corresponding strain gage measurements[4] (see Supplementary Fig. 1c) and yield $\Gamma_1 = 1.3$ J m$^{-2}$.

The secondary ruptures occur along the same interface as the first rupture, but the fracture energies, $\Gamma_i$ will vary widely as a result of the variation of the intervals, $\Delta t_i$, between the initiation of the $i - 1$ and the $i$th ruptures. $A(t)$, hence $\Gamma_i$ increase approximately logarithmically[5,44] with $\Delta t_i$. Since $\sigma_{yy}$ is constant throughout each rupture sequence, $\Gamma_i$ will scale with the contact drop $\delta A_i$ in the immediate wake of each rupture. Using the $i = 1$ rupture as a baseline, $\Gamma_i$ can be estimated via $\Gamma_i = \Gamma_1 \cdot (\delta A_i / \delta A_1)$. The fracture energies of the rupture sequence presented in Fig. 2 are listed in Supplementary Table 1.

## LEFM with a cohesive zone model

**Sub-Rayleigh ruptures.** Within the fracture mechanics framework, the propagation of rupture fronts is well-described by the LEFM solution for sub-Rayleigh shear cracks (Model II). The stresses in the vicinity of the rupture tip are predominantly described by

$$\sigma_{ij}(r,\theta) = \frac{K_{\mathrm{II}}}{\sqrt{2\pi r}} \Sigma_{ij}^{\mathrm{II}}(\theta, C_{\mathrm{f}}), \tag{1}$$

where $(r, \theta)$ are polar coordinated with respect to the rupture tip, $\Sigma_{ij}^{\mathrm{II}}(\theta, C_{\mathrm{f}})$ are known universal functions for the angular dependence of each component and the scalar $K_{\mathrm{II}}$ is the stress intensity factor[45]. This singular LEFM solution, which predicts infinite stresses at the rupture tip, is renormalized within a dissipative region located at the crack tip. This region, where the material strength is limited by $\tau_{\mathrm{p}}$, is commonly referred to as the cohesive zone or process zone[45].

As illustrated in Supplementary Fig. 1a, a direct measure of the characteristic size of the cohesive zone can be determined by the spatial scale, $X_{\mathrm{c}}$, over which the contact area drops at a rupture tip[4]. Examples of renormalized contact area in space for two typical ruptures, both rapid $0.99C_{\mathrm{R}}$ and slow $0.02C_{\mathrm{R}}$, are presented in Supplementary Fig. 1a. $X_{\mathrm{c}}$, decreases as $C_{\mathrm{f}} \to C_{\mathrm{R}}$ to zero, an effective 'Lorentz' contraction' described by fracture mechanics $X_c = X_c^0/f_{\mathrm{II}}(C_{\mathrm{f}})$, where $f_{\mathrm{II}}(C_{\mathrm{f}})$ is a known universal function[45]. The examples in Supplementary Fig. 1a demonstrate contraction, while Supplementary Fig. 1b demonstrates that this general result of elastodynamic theory (black line) describes the measurements (points) well.

We now make the assumption that $X_{\mathrm{c}}$ is independent of $\Gamma$, and is given by the above expression. Using the measured values of $\Gamma_i$ and $X_c^0$ and assuming shear stress weakening within the cohesive zone[47] as $\tau(x) = (\tau_{\mathrm{p}} - \tau_{\mathrm{r}}) \cdot \tilde{\tau}((x - x_{\mathrm{tip}})/X_c) + \tau_{\mathrm{r}}$, where $\tilde{\tau}(\xi) = e^\xi$, we can estimate the elusive constitutive parameters that define the dissipative processes and material properties under the extreme conditions near the rupture tip. The peak shear strength $\tau_{\mathrm{p}}$ can be directly inferred using the following expression[28,48]:

$$\Gamma_i = (\tau_p - \tau_r)^2 X_c \frac{\tilde{G}(C_{\mathrm{f}}, \tilde{\tau}(\xi))}{2\pi(1-k^2)\mu}, \tag{2}$$

where $\tilde{G}(C_{\mathrm{f}}, \tilde{\tau}(\xi)) = f_{\mathrm{II}}(C_{\mathrm{f}})[\int_{-\infty}^{0} \frac{\tilde{\tau}(\xi)}{\sqrt{-\xi}} d\xi]^2$. Here, $\tau_{\mathrm{r}}$ and $\mu$ are, respectively, the residual stress along the interface and shear modulus, $k = C_{\mathrm{S}}/C_{\mathrm{L}}$. Strain and stress can be obtained, as well as the slip function $u_{\mathrm{x}}$, by calculating explicitly the dynamic fields[28]. We note that this model differs slightly from that used in ref. 28, where ruptures propagated with different speeds but had a constant $\Gamma$ (or constant peak shear strength $\tau_{\mathrm{p}}$)[28]. Here the fracture energy (or peak shear strength) utilized is the value $\Gamma_i$ that is produced by the contact area drop of each secondary rupture. More details of the model can be found in ref. 28.

**Supershear ruptures.** Supershear ruptures are very prevalent secondary ruptures. In the case of supershear cracks, stress fields ahead of the tip are only coupled to the system's longitudinal waves. These create a singularity at the crack tip which can be described by $\sigma_{ij} \sim 1/r^{g}$[49]. The singular exponent $g$ depends strongly on the crack speed $(g(C_{\mathrm{f}}) \le 1/2)$, an effect notably different from sub-Rayleigh cracks. To compare the measured strain profiles with fracture mechanics predictions, we extend the sub-Rayleigh cohesive zone model directly to the supershear regime. The regularization of the shock wave singularity arises naturally when a cohesive zone is introduced, as it effectively regulates the stress divergences at the crack tip[28]. Details of the model can be found in ref. 28.

**Comparison between LEFM and experiments.** For comparison of LEFM to the strain gauge signals (Fig. 3d), we determine the strain variations, $\delta\varepsilon_{ij}$, due to each rupture passage by using the peak strain variations within a 10μs time window on either side of a rupture's passage. This window is necessary to compensate for any uncertainty in the precise arrival time of each (weak) rupture, as these variations mainly result[4] from the angular function $\Sigma_{ij}^{\mathrm{II}}(\theta, C_{\mathrm{f}})$.

## Undetected ruptures

The roll-off at the left-top as shown in the inset of Fig. 4b may be an indication that ruptures with slips smaller than 3 μm are only partially detected. For example, in the interval between the 8th to 9th ruptures that were labeled in Fig. 2b, it is possible to observed a number of additional shallow ruptures. During this period, the contact area does not increase monotonically (as would be expected from aging after the cessation of slip). $A(x, t)$ is, instead, essentially flat with small fluctuations. This indicates a weakening of the contact interface induced by secondary ruptures which are nearly below our detection threshold.

## Estimation of the seismic moment for secondary ruptures

When calculating the seismic moment, the two key inputs are the mean slip and the area within a fault where slip takes place. In our experiment, it is challenging to precisely calculate the seismic moment: $M_{0(i)} = \int_0^L GW \delta u_{x(i)}(x)dl$, since we do not know the slip distribution along the whole interface. To estimate $M_{0(i)}$ we simplified the calculation by using $M_{0(i)} \approx G \cdot W \cdot \delta u_i L = G \cdot S \cdot \delta u_i$, where $S = WL$ the entire interface area (since the secondary ruptures generally traverse the entire interface). With this simplification, the seismic moments of the secondary ruptures vary from 1 to 50 N·m. The magnitudes of these lab-earthquakes are, therefore, close to − 5, based on the formula $2/3(\log M_0 - 9.1)$[50].

## Data availability

The experimental data[51] generated in this study have been deposited in the Zenodo database under access code https://doi.org/10.5281/zenodo.10073589.

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

## Acknowledgements

We thank M. Adda-Bedia (ENS Lyon) and D. Kammer (ETH Zurich) for helpful discussions. This work was supported by Israel Science Foundation, grant numbers 840/19.and 416/23. S.S. acknowledges the support of the Emily Erskine Endowment Fund. M.W. acknowledges the support of the Lady Davis Fellowship Trust.

## Author contributions

S.S. and J.F. conceived of the project. S.S. designed and performed research with help of M.W and Y.P. S.S and J.F. analyzed the results. S.S. and J.F. wrote the manuscript. J.F. supervised the research. All authors discussed the results and commented on the manuscript.

## Competing interests

The authors declare no competing interests.
