## [Peer Review File · Nature Communications]

How frictional slip evolvesREVIEWER COMMENTS

Reviewer #1 (Remarks to the Author):

The authors present a comprehensive study on frictional slip. The set-up is constituted by two PVC blocks normally and tangentially loaded. Contact area high frequency visualization during crack propagation shows many secondary cracks which, contrary to expectations, account for the majority of the interfacial slip. To the best of my knowledge this result is fully original and high valuable. I recommend publication of the present paper after minor revisions.

In Fig. 1 the reader understands that there are grooves at the interface. Nevertheless this is explained only in the "Methods" section, and I could not judge the impact of this on the whole system. I suggest to include this information in the main text and to explain to the reader the role that these grooves may play in the whole propagation of the crack. Are they "stopping" the crack during its propagation?

As slip advances discontinuously also dissipation does. Is there anything we can learn? For example shall we expect a non-uniform damage along the interface? Or contact area aging then "cancel-out" all the interface anisotropy that eventually slip has created with multiple ruptures?

Reviewer #2 (Remarks to the Author):

This is an interesting and mostly well-written manuscript about frictional slip. In short, high-resolution experimental measurements allowed the authors to establish that after the initial rupture initiating each stick-slip event, a series of secondary ruptures occur. This is in contrast to the traditional viewpoint of continuous motion or "dynamic friction" assumed to take place after the transition from static to dynamic friction after the initial rupture. I have some question and comments I believe the authors should address:

(i) The idea that individual slip events of stick-slip motion consist of a sequence of many "micro-slip" events is not new; see, e.g., the work of Santucci and coworkers on peeling of an adhesive tape (Phys. Rev. Lett. 115, 128301). I think the authors should put their work better in the context of such closely related works, and comment on the similarities and differences between them and their results. This also relates to the question about the novelty of the present work: I think more transparency is needed on what aspects of the presented results are genuinely novel, and if some parts of the story constitute merely a re-application of existing ideas from the literature in a somewhat new context.

(ii) The main result here is that what is usually understood to be "dynamics friction" is in fact a sequence of secondary ruptures. While this is certainly interesting, I wonder if this observation can be put into more practical use by, e.g., finding a new way to compute the dynamic friction coefficient by coarse-graining the microscopic dynamics? The authors seem to mention something like this when they write that "global measurements that average over spatial and significant temporal scales present a mean-field perspective of the average slip", but the related discussion does not seem to be very conclusive nor detailed.

(iii) At the end of the manuscript, the authors write "The complex framework of inhomogeneous and coexisting sequenced ruptures are statistically observed as aftershocks of large earthquakes within natural faults." I have some problems in following the meaning and interpretation of this statement. Are the authors trying to argue that the secondary ruptures they observe in their lab-scale friction experiment would somehow be analogous to aftershocks in real earthquakes? If so, what is the evidence for that? For instance, do the secondary ruptures obey something like the Omori law, and/or other empirical statistical laws aftershocks of real earthquakes are found to obey?

(iv) While the paper is mostly well-written, the first sentence of the abstract (!) does not seem to

make sense, possibly because something is missing from it, or because there is an extra "that":
"Earthquake-like ruptures that break the contacts that form the frictional interface separating contacting bodies and mediate the onset of frictional motion (stick-slip)."

We would like to express our gratitude for the insightful comments and constructive suggestions provided by both referees. These have significantly enhanced the clarity and accuracy of our manuscript. We believe that the modifications made, which were based on the referees' suggestions, have significantly improved the manuscript.

We have prepared the revised paper with red marks for any changes. To address the reviewers' comments, a number of modifications were made to the main manuscript and, in addition, we have added two new figures (Supplementary Fig. 2 and Supplementary Fig. 6). A new reference has also been added.

Below we include our point-by-point response to the referees' comments, with their comments in black, our responses in blue and resulting additions made to the manuscript in red. We believe that all of the referee's concerns have been satisfactorily addressed.

Reviewer: 1

Referee: In Fig. 1 the reader understands that there are grooves at the interface. Nevertheless this is explained only in the "Methods" section, and I could not judge the impact of this on the whole system. I suggest to include this information in the main text and to explain to the reader the role that these grooves may play in the whole propagation of the crack. Are they "stopping" the crack during its propagation?

Response: Thanks for this suggestion. We now note the influence of the grooves on the rupture dynamics explicitly within the main text. We find that grooves have little effect on the rupture propagation. To demonstrate this, we now provide a new supplementary figure (Supplementary Fig. 2a) which illustrates that front velocities are not sensitive to the grooves imprinted on the interface. Furthermore, the strain measurements and their correspondence to LEFM predictions, as well as the value of the fracture energy are not affected by the existence of the grooves. This is now demonstrated in Supplementary Fig. 1c at $X = 100$ mm (strain measurements above the grooves) and Supplementary Fig. 2b at $X = 140$ mm (strain measurements without grooves). Comparison of the two plots shows that the fracture energy is not affected by the existence of the grooves. The strain field at the mesoscopic scale of the grooves is independent of the local topology, even in the case of the grooves, where the microscopic asperities are patterned instead of randomly distributed. The grooves indeed redistribute the stress at the micro-scale within the interface. This redistribution increases the local contact density at the top of grooves (while eliminating contacts within the grooves), but it influences neither the nominal stress distribution at the macro-scale (e.g. millimeters) nor the local propagation of the ruptures.

To clarify these issues, we have added a few sentences within the main text (line 51):

'... the displacements $u_x^+(x, t)$ and $u_x^-(x, t)$ of the upper and lower blocks at 30 spatial points within the central 3.6 mm of the interface. These displacements were measured by correlating the profiles of grooves engraved on both sides of the interface (Methods and Fig. 1d). These micro-scale grooves have negligible influence on the rupture propagation, resulting strain fields and the value of the fracture energy, as demonstrated in Supplementary Figs. 1&2 (Methods). Slip $u_x(x, t) = u_x^+(x, t) - u_x^-(x, t)$, was obtained at each point at a rate of 560,000/sec with a $0.5 \mu\text{m}$ resolution (Methods)....'

We further emphasized this point in the revised manuscript by adding the following to the Methods section (line 420):

'...The relative slip, $u = u^+ - u^-$ was thereby obtained with a resolution of better than $0.5 \mu\text{m}$ as presented in Fig. 1e (blue u^+ and red u^- curve). We note that the grooves produce a negligible effect on the rupture propagation, as shown in Supplementary Fig. 2a. Furthermore, the strain measurements and their correspondence to LEFM at $X = 100$ mm (with grooves, Supplementary Fig. 1c) and $X = 140$ mm (without

grooves, Supplementary Fig. 2b) show that both the grooved and non-grooved locations have the same fracture energy. While the grooves redistribute the contact area at the micro-scale within the interface (e.g. the local contact density increases at groove edges and is zero within the grooves), this does not influence the macro-scale rupture properties (i.e. the nominal stress distribution millimeters away from the interface or the propagation speeds).’

The new figure in the supplementary (Supplementary Fig. 2) reads:

Supplementary Fig. 2. Evaluation of the influence of interface grooves. **a**, An example of the front velocity $C_f(X)$ of the first rupture along an interface (corresponding to the experiment shown in Fig. 1c in the main text). The gray shadow indicates the grooved region. The front velocity increased rapidly at the initial stage and then stabilized when approaching $0.9C_R$. It is clear that the grooves do not noticeably perturb the propagation process. **b**, The measured strain components at $X = 140$ mm (area without grooves) and the LEFM-predicted form (when coupled to the cohesive zone model described in Methods). We note that this fit was obtained with the *same* value $\Gamma = 1.3$ J/m² as used at $X = 100$ mm (located above the grooved section), as shown in Supplementary Fig. 1c. The differences between the strains here and those in Supplementary Fig. 1c result from the slightly different values of C_f at the $X = 100$ mm and the $X = 140$ mm locations.

Referee: As slip advances discontinuously also dissipation does. Is there anything we can learn? For example shall we expect a non-uniform damage along the interface? Or contact area aging then "cancel-out" all the interface anisotropy that eventually slip has created with multiple ruptures?

Response: This is a good question. The dissipation and damage are indicated in these experiments by the value of the real contact area, $A(x, t)$. $A(x, t)$ varies with the rupture propagation history. The accompanying slip history introduces non-uniform residual strengths (and stresses) along the interface. Upon the cessation of slip at a given location, the strength of the interface will increase with time due to contact area aging. We find that this healing process is local and does not seem to ‘homogenize’ the local interface strength.

To show this, we have added a new figure, Supplementary Fig. 6, to the supplementary materials. The figure provides an example of the contact area variations at both at the ms scale and long-time scales (10’s of seconds). The figure shows that the contact (normalized before the initiation of the first rupture) drops to different levels along the interface after the passage of the primary and secondary ruptures. Once slip stops, the contact area indeed ages at each point; increasing with the same aging law (that is logarithmic at long scales). The recovery of the strength at each spatial location depends on the local values of the residual strength at the cessation of slip, aging time and applied stress levels. At the cessation of slip the residual interface strengths vary along the interface, so that, even at long times, the interface strength does not homogenize in space. The contact areas at each spatial location indeed ‘remember’ their initial values.

To clarify this point, we now add the following discussion about interface dissipation and damage to the main text (line 303) as follows:

‘...as most of u_{tot} is directly attributed (Fig 4) to the rupture process. The dissipation and ‘damage’ (damage in these experiments is measured by the variation of the local contact area) imposed by the primary and secondary rupture processes are non-uniform along the interface. These ruptures, with their accompanying slip, introduce non-uniform residual strengths along the interface. This non-uniformity is retained upon the cessation of slip. While the interface strength at each point will heal via the approximate logarithmic aging in time⁵ of A (Supplementary Fig. 6), any inhomogeneity of A , upon the cessation of slip, will be retained by this healing process. $A(x, t)$ depend on the initial values of the residual strength at the cessation of slip, local stress levels and the time since slip cessation *at each spatial point*.’

To demonstrate this, we have added Supplementary Fig. 6:

Supplementary Fig. 6. The contact area evolution along the interface at different time scales. a. 3D map of the contact area changes over the first 4.5 ms. $A(x, t)$ is normalized prior to the first (primary) rupture. **b.** 3D map of the contact area changes in the 50 seconds following rupture propagation. The ruptures create a non-uniform contact area drop and then heal via contact aging. The healing does not ‘homogenize’ the contact area changes along the interface. **c.** Logarithmic aging of the contact area after the cessation of slip (A/A_{res} , where A_{res} is the residual contact area immediately upon the passage of all secondary ruptures) at various spatial positions ($X \in [50, 150]$ mm) from **b**. The inset image is plotted in logarithmic time. The long-time aging rates are identical at each location.

Reviewer: 2

Referee: (i) The idea that individual slip events of stick-slip motion consist of a sequence of many "micro-slip" events is not new; see, e.g., the work of Santucci and coworkers on peeling of an adhesive tape (Phys. Rev. Lett. 115, 128301). I think the authors should put their work better in the context of such closely related works, and comment on the similarities and differences between them and their results. This also relates to the question about the novelty of the present work: I think more transparency is needed on what aspects of the presented results are genuinely novel, and if some parts of the story constitute merely a re-application of existing ideas from the literature in a somewhat new context.

Response: The work of Santucci and coworkers on the peeling of an adhesive tape (Phys. Rev. Lett. 115, 128301) indeed reveals that the peeling of the tape is dominated by the effects of a previously undetected rupture mode that produces peeling 'steps' that, when observed at large time scales, produce macroscopic slip that appears to be continuous. The reviewer is correct that these effects are, indeed, in a qualitative sense, analogous to the effects of the secondary ruptures described in this paper. We thank the reviewer for this comment, as the qualitative similarity of our results with adhesive peeling dynamics had escaped our notice. The detailed behaviors and the overall physics driving these two systems (peeling vs frictional motion) are, of course, quite different, but the resultant large-scale dynamics have, indeed, a quite similar appearance. Some of the key differences include: the mode of fracture (roughly tensile in peeling vs shear-driven in friction), the propagation mode of the excited secondary ruptures (transverse to the macroscopic slip direction in peeling, parallel to macroscopic slip in friction), and critically, the fact that the fracture energy is highly history-dependent in the case of friction (due to the extremely short and erratic healing times between ruptures – rendering them nearly 'invisible') whereas, in peeling, the secondary ruptures 'fracture' only fresh adhesive layers.

We now note the similarity of both peeling and friction in the discussion section (line 288) stating:

‘... Global measurements that average over spatial and significant temporal scales present a mean-field perspective of the average slip. These would manifest as the smooth curve assumed in empirical friction laws^{7,23}. **Such apparently continuous macroscopic motion that is, instead, driven by a sequence of small discrete events is also observed in the peeling dynamics of adhesive tape⁴⁵. While the details are different than in the frictional system, the apparent slow and continuous macroscopic motion obtained when peeling an adhesive strip is actually produced by a discrete succession of nearly imperceptible rapid fronts, each producing a small and discrete amount of slip.**’

Referee: (ii) The main result here is that what is usually understood to be "dynamics friction" is in fact a sequence of secondary ruptures. While this is certainly interesting, I wonder if this observation can be put into more practical use by, e.g., finding a new way to compute the dynamic friction coefficient by coarse-graining the microscopic dynamics? The authors seem to mention something like this when they write that "global measurements that average over spatial and significant temporal scales present a mean-field perspective of the average slip", but the related discussion does not seem to be very conclusive nor detailed.

Response: It is certainly a good idea to try to utilize the secondary ruptures to, in a sense, control frictional behavior. In this paper we have, for the first time, demonstrated their existence. Additionally we have described the characteristics of these ruptures as well as their contribution to frictional slip. The fundamental understanding that the majority of the slip in frictional dynamics is mediated by this, previously unexplored, phenomenon provides a new way to view frictional motion (much in the way that the work of Dalbe et al demonstrated that the dynamics are peeling were far from understood). As the reviewer suggests, the fundamental understanding obtained here could possibly, in the future, be utilized to control different aspects of frictional motion. For example, artificial nucleation or excitation of secondary ruptures could, in principle, enhance the amount of frictional

slip per macroscopic slip event. On the other hand, insertion of rupture barriers within a frictional interface could, conceivably, be used to suppress macroscopic frictional slip (by arresting secondary ruptures and therefore enabling more effective healing of an interface). This is certainly a very fruitful direction to explore in future work. Practical utilization of this new fundamental understanding of the origin of frictional slip is, however, beyond the scope of the current paper.

We now briefly note the above (line 294) stating:

‘Now that the basic physics of secondary frictional ruptures are understood, we speculate that it may be possible to utilize this knowledge to control the amount of frictional slip that takes place in a given event. In principle, external excitation of numerous secondary ruptures should enhance overall frictional slip, while the insertion of barriers to rupture³ could be a means to limit slip by arresting (trapping) secondary ruptures.’

Referee: (iii) At the end of the manuscript, the authors write "The complex framework of inhomogeneous and coexisting sequenced ruptures are statistically observed as aftershocks of large earthquakes within natural faults." I have some problems in following the meaning and interpretation of this statement. Are the authors trying to argue that the secondary ruptures they observe in their lab-scale friction experiment would somehow be analogous to aftershocks in real earthquakes? If so, what is the evidence for that? For instance, do the secondary ruptures obey something like the Omori law, and/or other empirical statistical laws aftershocks of real earthquakes are found to obey?

Response: Since the work of Byerlee in late 1960's and early 1970's, the close relation between earthquakes and friction has been apparent. On the other hand, both earthquakes and friction, as the current work demonstrates, are quite far from being fundamentally understood. Here we have shown that extended sequences of secondary ruptures are triggered after each 'main event'. It is, therefore, not a far stretch to assume that such secondary ruptures are also excited after a large earthquake that takes place within natural faults. The largest of these ruptures would conceivably be identified as the 'aftershocks' that follow large earthquakes. As we demonstrated in the paper, the slip of each secondary rupture can be related to its effective moment. Indeed, in Fig. 4b, we showed that the statistics of secondary slip magnitudes are, in fact, quite similar to the Gutenberg-Richter (G-R) law that describes the magnitude distribution for natural earthquakes. A new insight suggested by this analysis is that earthquake magnitudes (hence G-R statistics) may be dominated by the slip distributions of events and not, as often assumed, by earthquake propagation lengths. This is noted explicitly in the text (line 252):

‘Our results may suggest that the main reason for the spread of M_0 is *not* variations of the spatial extent of the different events, since the ruptures observed here all roughly span the interface length. Instead, M_0 is governed by δu_i . δu_i are mainly controlled by the fracture energy, Γ_i , of each rupture, which, itself is governed by the healing time Δt_i . This could also be true for aftershock moments within natural faults⁴³.’

The reviewer's suggestion to compare secondary rupture statistics to the Omori law is a good one. This comparison is, however, difficult to achieve in our current experimental system. The Omori law describes how the frequency of aftershocks decreases with time after the main shock. The signal to noise ratio of our slip measurements, however, is too low to enable the detection of a sufficiently large number of small events to enable a reliable statistical analysis of all of the ruptures excited after each main event. In addition, the slip measurements are only performed in a small section of the interface—many small (spatially confined) events may take place outside of this measurement section. As a result, measurement of Omori-type statistics is better left to acoustical analysis of laboratory earthquakes, such as that described by Goebel, T.H., Brodsky, E.E. and Dresen, G., (Geophys. Res. Lett. 50, e2022GL101241).

In response to the referee's comment, we have now modified our discussion of the relation of

secondary slip sequences and aftershocks within natural faults to clarify the parallels that we believe exist (line 316):

‘... These myriad events are so numerous (and so small) that they are individually invisible and are generally only represented by statistical tools (e.g. Gutenberg-Richter analysis). The secondary ruptures that we observe are aftershocks, each is part of a swarm of rupture events in the immediate aftermath of any ‘main’ stick-slip event. These rupture sequences could provide a means to relate observed correlations between geodetically observed afterslip and seismically observed aftershocks^{10,11}. These experiments provide valuable insights to complex fault processes, as detailed observation of individual ruptures within natural faults is currently extremely challenging.’

Referee: (iv) While the paper is mostly well-written, the first sentence of the abstract (!) does not seem to make sense, possibly because something is missing from it, or because there is an extra "that": "Earthquake-like ruptures that break the contacts that form the frictional interface separating contacting bodies and mediate the onset of frictional motion (stick-slip)."

Response: We thank the reviewer for her/his careful reading. It is true there is an extra “that”. We corrected it as:

‘Earthquake-like ruptures break the contacts that form the frictional interface separating contacting bodies and mediate the onset of frictional motion (stick-slip).’

REVIEWERS' COMMENTS

Reviewer #1 (Remarks to the Author):

The authors have consistently improved the manuscript and the response to my comments is satisfactorily. I suggest publication in the present form.

Reviewer #2 (Remarks to the Author):

My take is that the authors have addressed my comments in an appropriate fashion, and I hence recommend publication of the paper in its current form.